# Metabolic and Photosynthesis Analysis of Compound-Material-Mediated Saline and Alkaline Stress Tolerance in Cotton Leaves

**DOI:** 10.3390/plants14030394

**Published:** 2025-01-28

**Authors:** Mengjie An, Yongqi Zhu, Doudou Chang, Xiaoli Wang, Kaiyong Wang

**Affiliations:** 1Xinjiang Key Laboratory of Biological Resources and Genetic Engineering, College of Life Science & Technology, Xinjiang University, Urumqi 830046, China; amj10@xju.edu.cn (M.A.); yongqizhu@xju.edu.cn (Y.Z.); 2The Key Laboratory of Oasis Eco-Agriculture, Xinjiang Production and Construction Corps/Agricultural College, Shihezi University, Shihezi 832000, China; changdd0624@163.com; 3Xinjiang Agricultural Vocational Technical College, Changji 831100, China

**Keywords:** saline stress, alkaline stress, ABC transporters, arginine biosynthesis, C5-branched dibasic acid metabolism

## Abstract

Soil salinization and alkalization can cause great losses to agricultural production in arid regions. Cotton, a common crop in arid and semi-arid regions in China, often encounters saline stress and alkaline stress. In this study, NaCl (8 g·kg^−1^), Na_2_CO_3_ (8 g·kg^−1)^, and a compound material (an organic polymer compound material) were mixed with field soil before cotton sowing, and the ion content, photosynthetic characteristics, and metabolite levels of the new cotton leaves were analyzed at the flowering and boll-forming stage, aiming to clarify the photosynthetic and metabolic mechanisms by which compound material regulates cotton’s tolerance to saline stress and alkaline stress. The results showed that the application of the compound material led to an increase in the K^+^/Na^+^ ratio, stomatal conductance (Gs), efficiency of PSII photochemistry (ψPSⅡ), potential activity (Fv/Fo), and chlorophyll content (Chla and Chlb), as well as the abundances of D-xylonic acid and DL-phenylalanine in the NaCl treatments. Additionally, there were increases in the K^+^ content, K^+^/Na^+^ ratio, Chla/b ratio, net photosynthetic rate (Pn), transpiration rate (Tr), ψPSⅡ, and D-saccharic acid abundance in the Na_2_CO_3_ treatments. A correlation analysis and a metabolic pathway analysis revealed that the compound material mainly regulated the photosynthetic characteristics of and the ion balance in the new leaves through regulating the abundance of key metabolites when the cotton was under NaCl stress or Na_2_CO_3_ stress. Furthermore, the positive impact of the compound material on the cotton’s NaCl stress tolerance was stronger than that on the cotton’s Na_2_CO_3_ stress tolerance.

## 1. Introduction

Soil salinization, a serious environmental problem, often causes saline and alkaline stresses to crop growth [1]. Salinized soils contain a large amount of sodium salts, of which, the neutral salts NaCl and Na_2_SO_4_ and the alkaline salts NaHCO_3_ and Na_2_CO_3_ account for a large proportion. Among them, many previous studies have used NaCl to simulate neutral salt and Na_2_CO_3_ to simulate alkaline salt [2]. Neutral salts and alkaline salts have different impacts on crop growth [3]. Neutral salt stress often causes ion damage and osmotic stress in plants, while alkaline salt stress not only causes the above damage to plants but also increases the pH of plants [4].

Cotton has a certain degree of salt tolerance. However, high concentrations of sodium salts in the soil could still reduce the emergence rate and the growth rate of cotton, resulting in a cotton yield loss [5]. Roy et al. [6] reported that cotton responded to saline and alkaline stresses through some metabolic changes related to photosynthesis, ion transport, hormone synthesis, osmotic adjustment, and solute accumulation. Many saline and alkaline response pathways, such as the energy metabolism pathway and the osmotic-regulation pathway, have been found to play key roles in the saline and alkaline tolerance of many crops [7,8]. High concentrations of sodium salt in soil reduce the chlorophyll production and photosystem II efficiency of crop leaves and cause metabolic dysfunction and the inhibition of photosynthesis and other physiological and biochemical activities, leading to growth arrest and even death [9,10]. Therefore, it is necessary to conduct a metabolic analysis to identify the key metabolites and metabolic pathways involved in the photosynthetic responses of cotton to high concentrations of sodium salt, which could provide a reference for regulating crop salt stress tolerance and increase the crop yield.

Exogenous compound material application is an effective way to alleviate sodium salt stress on crops. Nie et al. [11] found that the application of salicylic acid could improve the alkaline tolerance of cucumber seedlings through activating the reactive oxygen species-scavenging system, reducing Na^+^ toxicity, and improving chlorophyll content and photosynthetic capacity. Zhou et al. [12] showed that application of glutathione could decrease the Na^+^ content in tomato roots and leaves under NaCl stress and promote the selective transport of K^+^ and Ca^2+^ from roots to leaves, thus achieving ion balance and decreasing the damages caused by NaCl stress. Furthermore, the application of glutathione could also regulate the transformation of different forms of polyamines, promoting the synthesis and metabolism of polyamines. Shafiq et al. [13] showed that the application of copper oxide nanoparticles could alleviate salt stress on maize photosynthesis through regulating the N, P, K, Ca, and Mg contents in the roots and shoots. However, the crop cultures in the above studies were all conducted in the laboratory, and relevant studies based on field crop cultures of greater importance for agricultural production are scarce.

In our previous study [14], we discovered that the utilization of organic polymer compound material significantly enhances the activities of peroxidase (POD) and catalase (CAT), thereby effectively regulating the transport efficiency of potassium ions (K^+^) and sodium ions (Na^+^) in cotton leaves and roots to mitigate saline and alkaline stresses. Consequently, understanding the metabolic regulatory mechanism is of great significance. In this field barrel experiment, four treatments were designed, including the following: (1) CK-Y: NaCl was applied to soil; (2) P-Y: NaCl and a compound material were applied to soil; (3) CK-J: Na_2_CO_3_ was applied to soil; and (4) P-J: Na_2_CO_3_ and a compound material were applied to soil. Furthermore, the ion content, photosynthetic characteristics, and metabolite levels of the new cotton leaves were measured at the flowering and boll-forming stage after applying the compound material to the NaCl (8 g·kg^−1^) and Na_2_CO_3_ (8 g·kg^−1^) soils. The specific purposes of this study were (1) to compare the effects of the compound material on the photosynthetic characteristics and metabolites of cotton under saline and alkaline stresses (8 g·kg^−1^) and (2) to clarify the key metabolites and metabolic pathways involved in the regulatory mechanism of cotton’s photosynthesis responses to saline and alkaline stresses, mediated by the compound material. This study will provide a technical reference for improving the saline and alkaline stress tolerance of cotton through the application of a compound material in arid and semi-arid regions.

## 2. Results

### 2.1. Potassium and Sodium Ion Contents in New Cotton Leaves in Response to Saline Stress, Alkaline Stress, and Compound Material

The K^+^/Na^+^ ratio in the P-Y treatment was significantly higher, by 18%, than that in the CK-Y treatment. The Na^+^ content in the P-Y treatment was significantly lower, by 13%, than that in the CK-Y treatment. The K^+^ content and K^+^/Na^+^ ratio in the P-J treatment were significantly higher by 6% and 37%, respectively, than that in the CK-J treatment. The Na^+^ content in the P-J treatment was significantly lower, by 18%, than that in the CK-J treatment (*p* < 0.05) (Figure 1).

### 2.2. The Chlorophyll Content, Photosynthetic Gas Exchange Parameters, and Chlorophyll Fluorescence Parameters of New Cotton Leaves in Response to Saline Stress, Alkaline Stress, and Compound Material

The contents of Chla, Chlb, and Car in the new cotton leaves were increased by the compound material. The contents of Chla and Chlb in the P-Y treatment were significantly higher, by 37% and 37% (*p* < 0.05), respectively, than that in the CK-Y treatment. The Chla/b in the P-J treatment was significantly lower, by 10%, than that in the CK-J treatment (*p* < 0.05) (Figure 2A). The Gs in the P-Y treatment was significantly higher, by 49%, than that in the CK-Y treatment. The Pn, Gs, and Tr in the P-J treatment were significantly higher, by 21%, 20%, and 33%, respectively, than that in the CK-J treatment (*p* < 0.05) (Figure 2B). The Fv/Fo and the ψPSII in the P-Y treatment were significantly higher, by 9% and 6%, respectively, than that in the CK-Y treatment. The ψPSII in the P-J treatment was significantly higher, by 12%, than that in the CK-J treatment (*p* < 0.05) (Figure 2C).

### 2.3. Metabolome Analysis of New Cotton Leaves

A total of 1247 unique metabolites, such as lipids, lipid-like molecules, benzenoids, phenylpropanoids, polyketides, organoheterocyclic compounds, organic acids and their derivatives, and organic oxygen compounds, were detected (Figure 3A). The PCA results showed that the first principal component (PC1) explained 31.3% of the total variation, and the second principal component (PC2) explained 14.7% of the total variation (Figure 3B). There was a clear separation between the NaCl treatment and the Na_2_CO_3_ treatments by the PC2 and between the compound material treatments and the non-compound-material treatments by the PC1 (Figure 3B). In addition, a strong correlation was found between the replicates.

In the comparison of group CK-J vs. CK-Y, 164 metabolites were up-regulated, and 79 metabolites were down-regulated (Figure 4A). In P-Y vs. CK-Y, 69 metabolites were up-regulated and 21 metabolites were down-regulated. In P-J vs. CK-J, 73 metabolites were up-regulated and 59 metabolites were down-regulated (Figure 4A). In P-J vs. P-Y, 36 metabolites were up-regulated and 19 metabolites were down-regulated. The Venn diagram showed that there were ten differentially abundant metabolites (DAMs) for the four treatments (Figure 4B).

### 2.4. Key Metabolites Involved in the Response of Cotton Leaf Photosynthesis to Saline Stress, Alkaline Stress, and the Compound Material

With the criteria of VIP (variable importance in projection) being > 1 and 0.05 < *p* < 0.1, thirty-four DAMs were identified as the variables mostly relating to photosynthesis. These metabolites were classified as carbohydrates (16), amino acids (10), and fatty acids (8) (Table 1).

The abundances of D-xylonic acid (Log_2_FC = 1.34), DL-phenylalanine (Log_2_FC = 1.11), L-proline (Log_2_FC = 1.11), D-(-)-Glutamine (Log_2_FC = 1.30), portulacaxanthin I (Log_2_FC = 1.36), and L-phenylalanine (Log_2_FC = 1.31) in the P-Y treatment were higher than those in the CK-Y treatment. The abundance of D-raffinose (Log_2_FC = −2.83) in the P-Y treatment was lower than that in the CK-Y treatment. The abundances of D-saccharic acid (Log_2_FC = 1.34), eicosapentanoic acid (Log_2_FC = 1.04), and arachidonic acid (Log_2_FC = 1.29) in the P-J treatment were higher than those in the CK-J treatment. The abundance of isoleukotoxin diol (Log_2_FC = −1.86) in the P-J treatment was lower than that in the CK-J treatment (Table 1).

The VIP analysis results of the thirty-four DAMs (Figure 5) showed that for the NaCl treatments (CK-Y and P-Y), D-xylonic acid and DL-phenylalanine had high VIP scores, and their abundances in the P-Y treatment were higher than those in the CK-Y treatment. For the Na_2_CO_3_ treatments (CK-J and P-J), D-saccharic acid had a high VIP score, and its abundance in the P-J treatment was higher than that in the CK-J treatment. For the controls (the CK-J and CK-Y treatments), betaine had a high VIP score. For the compound material treatments (P-J and P-Y treatments), D-xylonic acid had a high VIP score.

The correlation analysis between the K^+^ content, Na^+^ content, photosynthesis, and key metabolites (Figure 6) showed that in the CK-Y treatment, DL-phenylalanine was negatively correlated with ψPSⅡ ((r (correlation coefficient) = −0.981) and qP (r = −0.988) (*p* < 0.01). In the P-Y treatment, D-xylonic acid was positively correlated with Gs (r = 0.978) (*p* < 0.01). DL-phenylalanine was negatively correlated with Ci (r = −0.989) and Tr (r = −0.963) (*p* < 0.01). In the CK-J treatment, betaine was negatively correlated with qP (r = −0.968) (*p* < 0.01). In the P-J treatment, DL-phenylalanine was negatively correlated with K^+^ (r = −0.960) (*p* < 0.01).

### 2.5. Abundance Levels of Key Metabolites Betaine, D-Xylonic Acid, DL-Phenylalanine, and D-Saccharic Acid in New Cotton Leaves in Response to Saline Stress, Alkaline Stress, and Compound Material

The analysis results of the abundance level changes of the four key metabolites under saline stress and alkaline stress (Figure 7) showed that the abundances of D-xylonic acid, DL-phenylalanine, D-saccharic acid, and betaine in the P-Y treatment were higher by 27%, 3%, 21%, and 2% (*p* < 0.05), respectively, than that in the CK-Y treatment. The abundances of D-xylonic acid, DL-phenylalanine, and D-saccharic acid in the P-J treatment were higher by 20%, 11% (*p* < 0.05), and 9%, respectively, than that in the CK-J treatment.

### 2.6. Key Metabolic Pathways Involved in Cotton Leaf Photosynthesis Under Saline Stress, Alkaline Stress, and Compound Material Treatment

ABC transporters, arginine biosynthesis, and C5-branched dibasic acid metabolism pathways were the key metabolic pathways involved in the response of saline- and alkaline-stressed cotton leaf photosynthesis to the compound material (Figure 8). D-raffinose, L-proline, L-phenylalanine, betaine, and L-glutamic acid were key metabolites of the ABC transporters pathway, L-glutamic acid and n-acetyl-glutamate were key metabolites of the arginine biosynthesis pathway, and L-glutamic acid and itaconate acid were key metabolites of the C5-branched dibasic acid metabolism pathway. The abundances of L-proline, L-phenylalanine, and betaine in the P-Y treatment were significantly higher by 4%, 6%, and 2%, respectively, than those in the CK-Y treatment (*p* < 0.05). The abundance of D-raffinose in the P-Y treatment was significantly lower, by 20%, than that in the CK-Y treatment (*p* < 0.05) (Figure 8).

## 3. Discussion

Under NaCl and Na_2_CO_3_ stresses, crops usually accumulate high concentrations of Na^+^ due to the competitive absorption of Na^+^ and K^+^ [15]. In this study, both the saline and the alkaline stresses had a significant negative effect on the ionic balance and physiological characteristics of the new cotton leaves. This is consistent with the study results of Liu and Su [16]. Furthermore, this study also found that the contents of Na^+^ and K^+^ in the new cotton leaves under the Na_2_CO_3_ stress were significantly higher than those under the NaCl stress. Therefore, the Na_2_CO_3_ stress had a greater effect on crops than the NaCl stress. This is due to the fact that saline stress generally leads to ionic damage and osmotic stress in plants. However, a high pH in alkaline stress directly affects plant roots’ growth and indirectly affects plants’ uptake of soil nutrients [17]. In addition to the Na^+^ and K^+^ contents, photosynthesis (gas exchange, chlorophyll fluorescence, and chlorophyll content) is also a physiological activity that is highly sensitive to saline and alkaline stresses [18], especially chlorophyll content [19]. Hamani, et al. [20] found that a 150 mM NaCl stress had a significant negative effect on the photosynthesis of cotton leaves by reducing gas exchange and chlorophyll fluorescence parameters. Kolomeichuk, et al. [21] found that NaCl stress could also reduce the Fv/Fo and Fv/Fm of crop leaves. In this study, it was found that the Pn, Tr, ψPSII, and qP of the new cotton leaves under the Na_2_CO_3_ stress were all significantly lower than those under the NaCl stress. This proved that the impact of the Na_2_CO_3_ stress on the photosynthesis of the cotton was greater than that of the NaCl stress.

The application of exogenous substances under saline and alkaline stresses can effectively alleviate stress-induced damage, reduce the Na^+^ content in plant organs, maintain ion balance, and improve plants’ salt tolerance [12,22]. In this study, the self-developed compound material was applied to saline and alkaline soils, which significantly reduced the Na^+^ content in the new cotton leaves and increased the K^+^ content and K^+^/Na^+^ ratio. In particular, this positive impact of the compound material on the Na_2_CO_3_-stressed cotton’s ion balance was stronger than that on the NaCl-stressed cotton’s ion balance. Furthermore, this study also found a difference in the photosynthesis between the saline stress condition and the alkaline stress condition after the application of the compound material. In this study, the positive impact of the compound material on cotton leaf photosynthesis was stronger under the NaCl stress than under the Na_2_CO_3_ stress, and the photosynthetic efficiency and the content of the photosynthetic pigment were significantly increased by the application of the compound material. Specifically, the Gs and Tr were increased, and the Ci was decreased. This indicates that the photosynthetic rate of cotton can be improved by the compound material through the regulation of the non-stomatal limitation [23]. Furthermore, the increases in the photosynthetic pigments in the cotton leaves may be due to the fact that the compound material protects chlorophyll from the damage of reactive oxygen species through increasing the content of carotenoids (Figure 6) [24]. qP reflects the openness of photosystem II [25]. Wang et al. [26] reported that under Na_2_CO_3_ stress, the qP and the ΦPSII were decreased, and ion transport was inhibited. In this study, the qP of NaCl- and Na_2_CO_3_-stressed new cotton leaves were increased following the application of the compound material. This indicates that the compound material application could increase the ψPSⅡ and the proportion of energy in the dark reaction of cotton leaves [27]. However, under the NaCl and Na_2_CO_3_ stresses, the Fv/Fo and Fv/Fm of the cotton leaves were not significantly increased. This indicates that the effect of the compound material on the regulation of photosynthetic pigments or the reduction/reoxidation of quinone compounds was not significant [28]. The improvement of photosynthetic characteristics is due to the excellent water retention capacity of the polymer materials in the compound material, which could decrease the water evaporation in saline and alkaline soils [29]. Furthermore, the Ca^2+^, Mn^2+^, Zn^2+^, and Fe^2+^ in the compound material are absorbed and retained into the molecular network [30], but they will be released slowly into the soil when the crops are under NaCl and Na_2_CO_3_ stresses, which could alleviate the damages to cotton caused by NaCl and Na_2_CO_3_ stresses.

The compound material had different effects on the new cotton leaves under the NaCl stress and the Na_2_CO_3_ stress. The metabolome analysis results showed that 90 and 132 DAMs were identified in the NaCl treatments (CK-Y vs. P-Y) and the Na_2_CO_3_ treatments (CK-J vs. P-J), respectively. This indicates that the application of the compound material caused more changes in the metabolites of the new cotton leaves under the Na_2_CO_3_ stress than under the NaCl stress. Furthermore, it was also found that D-xylonic acid, D-saccharic acid, betaine, and DL-phenylalanine were the key metabolites in the VIP analysis. Among which, betaine is a plant-derived osmoprotective compound [31]. Hamani et al. have shown that an increase in betaine can effectively improve the stomatal characteristics of cotton seedlings, thereby rapidly improving the gas exchange and chlorophyll fluorescence of cotton seedling under NaCl stress [20]. In this study, the abundance of betaine in the cotton leaves was higher under the Na_2_CO_3_ stress than under the NaCl stress. This may be due to the fact that Na_2_CO_3_ stress decreases the Tr, ψPSII, and qP, leading to an increase in betaine abundance (Figure 7). The different results may be due to the difference in the metabolites between the seedling stage and the flowering and boll stage. Han et al. [32] reported that amino acids, as the intermediates or end products of some metabolic pathways, participated in the signal transduction in the plant stress response, and amino acid metabolism in cotton leaves was affected to varying degrees under NaCl stress. In this study, under the NaCl stress, the photosynthetic characteristics (Gs, Chlb, ψPSII, and Fv/Fo) of the cotton leaves were improved through increasing the abundance of D-xylonic acid and DL-phenylalanine (Figure 6 and Figure 7). Under the Na_2_CO_3_ stress, the application of the compound material improved the Pn and Tr by increasing the abundance of D-saccharic acid (Figure 6 and Figure 7). By improving the amino acid metabolism of new cotton leaves, the compound material could change the abundance of carbohydrates. However, carbohydrates could increase the osmotic potential and reduce the damage caused by salt stress to crops [33]. Therefore, the compound material could improve the tolerance of cotton to NaCl stress and Na_2_CO_3_ stress by regulating the photosynthesis, ion balance, and metabolites in cotton leaves.

ABC transporters, arginine biosynthesis, and C5-branched dibasic acid metabolism were the key metabolic pathways involved in the impacts of the compound material on the saline- and alkaline-stressed new cotton leaves. Zhu et al. [34] reported that the ABC transporters metabolic pathway plays a key role in cotton’s response to abiotic stress. The function of ABC transporters is to use the energy generated by hydrolyzing ATP to drive the transport of various molecules, including sugars, amino acids, metal ions, peptides, proteins, hydrophobic compounds, and their metabolites. ABC transporters are involved in various physiological functions of organisms, such as maintaining the balance of osmotic pressure inside and outside cells, lipid transport, etc. [35]. ABC transporters also play an important role in the regulation of ion homeostasis under saline and alkaline stresses [36]. In this study, D-raffinose, L-proline, L-phenylalanine, betaine, and L-glutamic acid were the DAMs in the ABC transporters metabolic pathway under the saline and alkaline stresses. Among them, D-raffinose is involved in decreasing the damage to the PSII of leaves caused by environmental stress [37]. The abundance of D-raffinose was significantly decreased by the compound material under the saline stress and the alkaline stress in this study, but decreased more under the saline stress. This indicates that D-raffinose had a better effect on decreasing the damage to the PSII of the leaves under the saline stress. L-proline, L-phenylalanine, and L-glutamic acid are amino acids. The accumulation of amino acids could provide sufficient energy for ABC transporters to regulate ion homeostasis [38]. In this study, the abundance of L-proline (*p* < 0.05), L-phenylalanine (*p* < 0.05), and L-glutamic acid were increased by the compound material under saline stress but were decreased under the alkaline stress. This indicates that L-proline and L-phenylalanine, as the preferred substrates of transporters, participate in the regulation of Na^+^/K^+^ homeostasis and had a more obvious change in abundance under saline stress than under alkaline stress. Betaine can regulate the high osmotic pressure through its transport and accumulation by ABC transporters and play a protective role against saline stress [39]. In this study, the abundance of betaine was significantly increased by the compound material under the saline stress but was decreased under the alkaline stress (Figure 8). This may be due to the lower output of betaine by the ABC transporters pathway under the saline stress, indicating that the damage caused by the saline stress to leaves was less than that of the alkaline stress. In the C5-branched dibasic acid metabolism pathway, itaconate acid is an important metabolic pathway involved in the photosynthetic process of cyanobacteria [40]. However, previous studies did not report the response of itaconate acid in cotton to abiotic stress. In this study, the abundance of itaconate acid was significantly increased by the compound material under the saline stress but was decreased under the alkaline stress. This indicates that the leaf photosynthesis could be increased by the application of the compound material. This can further increase the abundance of itaconate acid and the tolerance of crops to saline and alkaline stresses [41]. Moreover, the effect of the compound material on the saline-stressed cotton was also stronger than that on alkaline-stressed cotton. Wei et al. [42] found that the arginine biosynthesis pathway was involved in the response of cotton to saline stress; furthermore, N-acetyl-glutamate, a protein localizing chlorophyll, changed the abundance of the metabolite N-acetyl-glutamate by affecting arginine in photosynthesis. In this study, the N-acetyl-glutamate abundance was increased by the compound material under the saline stress and the alkaline stress. This may be due to the alleviation of the damages caused by saline stress and alkaline stress to cotton leaf chlorophyll abundance and photosynthesis by changing the abundance of N-acetyl-glutamate and the promotion of arginine biosynthesis by the compound material.

## 4. Materials and Methods

This experiment was performed at the Experimental Station of Grape Research Institute in Shihezi City, Xinjiang Uygur Autonomous Region, China (44°20′ N, 86°03′ E), from April 20th to September 20th. The experimental station experiences a temperate arid climate characterized by an average annual rainfall of 210 mm and an average annual evaporation rate of 1660 mm. The soil texture in the study area was loam. The soil’s physical and chemical properties, measured before the experiment, were as follows: The soil pH was 8.45. The soil salinity (EC1: 5) was 0.03 ds·m^−1^. The soil organic matter content was 12.5 g·kg^−1^. The alkali hydrolyzable nitrogen content was 54.3 mg·kg^−1^. The available phosphorus content was 11.7 mg·kg^−1^. The available potassium content was 218.5 mg·kg^−1^. The cotton variety Xinluzao 62 (*Gossypium hirsutum* L.) was used. The compound material (organic polymer compound material) was a mixture of calcium lignosulfonate, manganese sulfate, zinc sulfate, ferric sulfate, boric acid, anionic polyacrylamide, and polyvinyl alcohol (mass ratio: 4:4:4:2:1:0.5:0.5) [43]. It was a dark brown liquid, with a strong dispersing capacity, bonding capacity, and chelating capacity. Its pH was 2.51, and its EC was 21.1 ds·m^−1^ (patent number: AU 2020/200435 B1).

It can be seen from Figure 9A that the compound material was prepared through rapid stirring at a high temperature (50–130 °C). The composition of the compound material was complex, the peak absorptions at 3219 cm^−1^ and 1620 cm^−1^ could be attributed to the N-H stretching vibration and the bending vibration of the primary amine, respectively, the peak absorption at 1466 cm^−1^ could be attributed to the stretching vibration of the aromatic ring C=C, the peak absorption at 1145 cm^−1^ could be attributed to the symmetrical stretching vibration of PO_2_ in the inorganic phosphate, the peak absorption at 823 cm^−1^ could be attributed to the =C-H non-planar bending vibration on the benzene ring, and the peak absorption at 545 cm^−1^ could be attributed to the vibration of metal oxides (Figure 9B).

### 4.1. Plant Growth

This experiment adopted a randomized complete block design with four treatments, including the following: (1) CK-Y: NaCl was applied to the soil; (2) P-Y: NaCl and the compound material were applied to the soil; (3) CK-J: Na_2_CO_3_ was applied to the soil; and (4) P-J: Na_2_CO_3_ and the compound material were applied to the soil. Each treatment had three repetitions/barrels. On 20 April, cotton field soil was collected in layers and transferred into plastic barrels (0.5 m in diameter and 0.6 m in height), while keeping the original soil layer arrangement and bulk density. Then, the barrels were buried back into a cotton field. For the CK-Y and P-Y treatments, NaCl (8 g·kg^−1^) was thoroughly mixed with the plough layer (0–20 cm) in the barrels, while for the CK-J and P-J treatments, Na_2_CO_3_ (8 g·kg^−1^) was thoroughly mixed with the plough layer (0–20 cm) in the barrels. The soil pH and electrical conductivity (EC) (water:soil = 2.5:1) were 8.24 and 18.4 ds·m^−1^, respectively, for the NaCl treatments and were 9.78 and 10.3 ds·m^−1^, respectively, for the Na_2_CO_3_ treatments. On April 29th, 360 kg·hm^−2^ of urea (N: 46%) and 795 kg·hm^−2^ of compound fertilizer (17-17-17) were dissolved in water and applied, for all the treatments, through drip irrigation. The cotton was sown on May 4th. Six seedlings were retained in each barrel after cotton emergence. On May 6th, 300 kg·hm^−2^ of the compound material was dissolved in water and applied to the soils of the P-Y and P-J treatments. No fertilizers or compound material were applied thereafter. Irrigation was performed for the first time on June 25th, and then irrigation was conducted every 3 days. The total irrigation volume during the entire growth period was 4500 m^3^·hm^−2^. Other agricultural management processes, such as weeding, were consistent with those in local fields. At the flowering and boll-forming stage (August 19th), five new leaves (top leaves) with no diseases and pests were selected from each treatment for the determination of photosynthetic gas exchange and chlorophyll fluorescence parameters. Then, six new leaves were collected from each treatment. They were immediately placed in liquid nitrogen and stored at −80 °C for the untargeted metabolomic analysis.

### 4.2. Determination of Na^+^ and K^+^ Content

The leaf samples were immersed in 98% H_2_SO_4_ and 30% H_2_O_2_, and the contents of K^+^ and Na^+^ were determined using a flame spectrophotometer (AP1200 type, Shanghai, China), following the method of Bao [44].

### 4.3. Determination of Chlorophyll Pigments

The photosynthetic pigments were extracted in 80% (*v*/*v*) acetone after cutting the fresh leaves (20 mg) into small pieces. The extracts were centrifuged, and the pigments were re-extracted using pellets. Finally, the absorbance was measured spectrophotometrically at 663.2, 646.5, and 470 nm, and the contents of the chlorophyll (Chla and Chlb) and carotenoids (Car) were calculated according to the methods of Lichtenthaler [45].

### 4.4. Determination of Photosynthetic Gas Exchange Parameters

The net photosynthetic rate (Pn), stomatal conductance (Gs), transpiration rate (Tr), and intracellular CO_2_ concentration (Ci) were determined using the LI-6400 portable photosynthesis system (LI-COR, Lincoln, NE), following the instructions of Liu et al. [46].

### 4.5. Determination of Chlorophyll Fluorescence Parameters

The maximum variable fluorescence in darkness (Fv), minimum fluorescence after dark adaptation (Fo), minimum fluorescence under light (Fo′), maximum fluorescence after dark adaptation (Fm), maximum fluorescence after light adaptation (Fm′), and steady-state fluorescence after light adaptation (Fs) were measured for 15 min using a PAM-2100 modulated chlorophyll fluorescence instrument (WALZ, Germany). The light intensity was 400 μmol m^2^·s^−1^, and the maximal light intensity was 8000 μmol m^2^·s^−1^. The chlorophyll fluorescence parameters (Equations (1)–(4)) were calculated using the methods of Baker et al. [47].Potential activity (Fv/Fo) = (Fm − Fo)/Fo(1)Maximal efficiency of PSII photochemistry (Fv/Fm) = (Fm − Fo)/Fm(2)Efficiency of PSII photochemistry (ψPSII) = (Fm′ − Fs)/Fm′(3)Photochemical quenching coefficient (qP) = (Fm′ − Fs)/(Fm′ − Fo)(4)

### 4.6. Untargeted Metabolomic Analysis

To investigate the metabolic responses of saline-stressed and alkaline-stressed cotton to compound material, an untargeted global metabolomic analysis of the new cotton leaf samples were conducted using ultra performance liquid chromatography/mass spectrometry (LC/MS). Each leaf sample (50 mg) was ground with liquid nitrogen, transferred into a 1.5 mL EP tube containing 200 μL of a pre-cooled mixture of methanol and water (3:1, *v*/*v*), vortexed, and placed overnight at 4 °C. After centrifuging at 4 °C for 15 min (13,000 rpm), the supernatants were collected and stored at −80 °C. For mass spectrometry (LC-MS) detection, the samples were re-dissolved using a 50 μL mixture of isopropanol, methanol, and water (1:1:2, *v*/*v*/*v*) and centrifuged, and the supernatants were collected for the detection. During the whole analysis, the samples were placed in an automatic sampler at 4 °C. The samples were separated by DIONEX UltiMate_3000 ultra-high performance liquid chromatography (UHPLC, America, Thermo Fisher Scientific, Waltham, MA, USA) using a C18 column. The QC samples were inserted into the sample queue to monitor and evaluate the stability of the system and the reliability of the experimental data. For each sample, electrospray ionization (ESI) was used for positive and negative mode detection. The samples were separated by UHPLC and analyzed by the Q-Exactive mass spectrometer (Q-Exactive, America, Thermo Fisher Scientific) [48].

The raw data were subjected to a peak alignment, retention time correction, and peak area extraction using the Compound Discoverer 3.0 program. The metabolite structure was identified using exact mass matching (<25 ppm) and secondary spectrum matching (MZcloud database). SIMCA-P 14.1 (Umetrics, Umea, Sweden) was used for pattern recognition, and the data were preprocessed by Pareto-scaling for the multi-dimensional statistical analysis.

### 4.7. Statistical Analysis

The metabolites with VIP > 1 and 0.05 < *p* <0.1 were selected as the DAMs. A Venn diagram was drawn to identify the common and specific DAMs. A one-way analysis of variance (ANOVA) was performed using the Duncan test (*p* < 0.05, SPSS 22.0). A correlation analysis was performed to obtain the Pearson correlation coefficient. All of the above analyses were performed in R software (Version 3.2.3, http://www.r-project.org) and Origin 8.0 software.

## 5. Conclusions

For the first time, we applied an organic polymer compound material to increase cotton’s resistance to saline and alkaline stresses and identified the key metabolites involved in the regulatory mechanism of cotton photosynthesis. The content of Na^+^ in new cotton leaves under the Na_2_CO_3_ stress was higher, while the photosynthesis rate was lower, compared with that of the cotton under the NaCl stress. Under the NaCl stress, the application of the compound material mainly increased the abundance of D-xylonic acid and DL-phenylalanine and improved the photosynthetic gas exchange, chlorophyll content, and chlorophyll fluorescence of the new cotton leaves. However, under the Na_2_CO_3_ stress, the application of the compound material mainly increased the abundance of D-saccharic acid and improved the photosynthesis of the new leaves. The compound material improved the cotton’s saline and alkaline stress tolerance through the regulation of the ABC transporters, arginine biosynthesis, and C5-branched dibasic acid metabolism metabolic pathways, photosynthesis, and Na^+^/K^+^ homeostasis. This study provides a technical reference for regulating the NaCl stress and Na_2_CO_3_ stress tolerance of cotton by using the compound material in arid and semi-arid regions.

## Figures and Tables

**Figure 1 plants-14-00394-f001:**
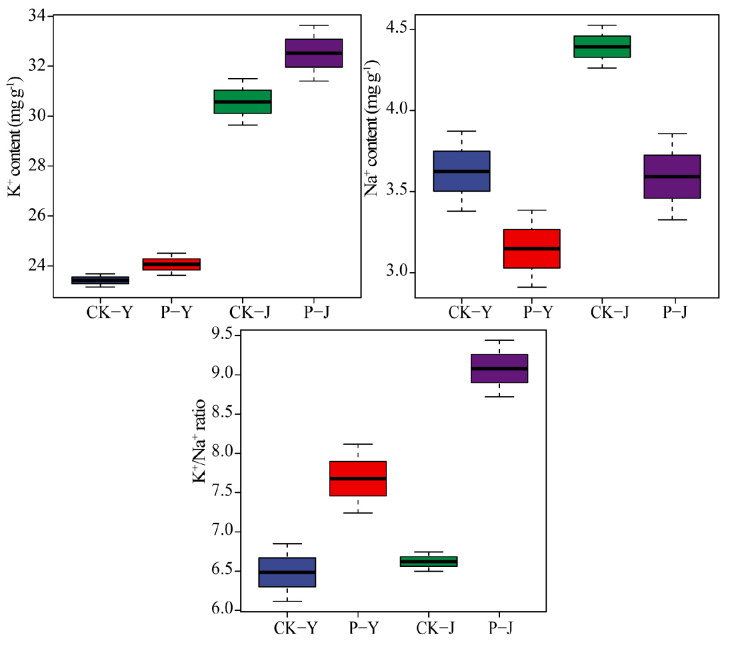
The effect of the application of a compound material on the contents of K^+^ and Na^+^, as well as K^+^/Na^+^, in leaves.

**Figure 2 plants-14-00394-f002:**
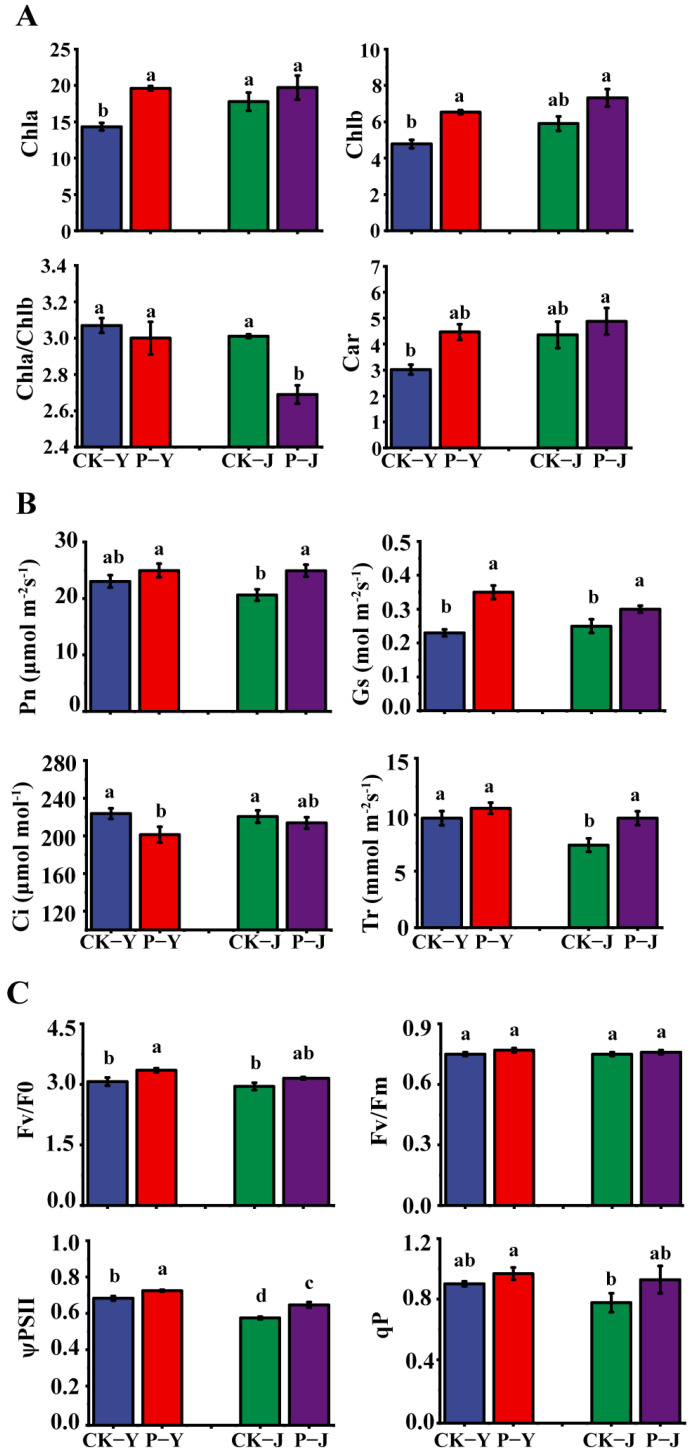
The differences in the photosynthesis of cotton leaves. (**A**) Photosynthetic pigment content; (**B**) photosynthetic performance; (**C**) chlorophyll fluorescence. Pn: net photosynthetic rate, Gs: stomatal conductance, Tr: transpiration rate, Ci: intracellular CO_2_ concentration, Fv/Fo: the potential activity, Fv/Fm: the maximal efficiency of PSII photochemistry, ψPSII: the efficiency of PSII photochemistry, qP: the photochemical quenching coefficient. Note: different letters in the bar chart indicate significant differences (*p* < 0.05); the same below.

**Figure 3 plants-14-00394-f003:**
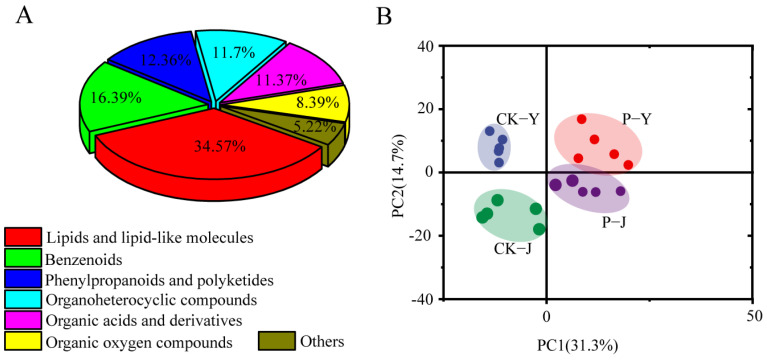
Metabolic summary analysis. (**A**) Principal component analysis (PCA) of metabolites; (**B**) classification of total metabolites.

**Figure 4 plants-14-00394-f004:**
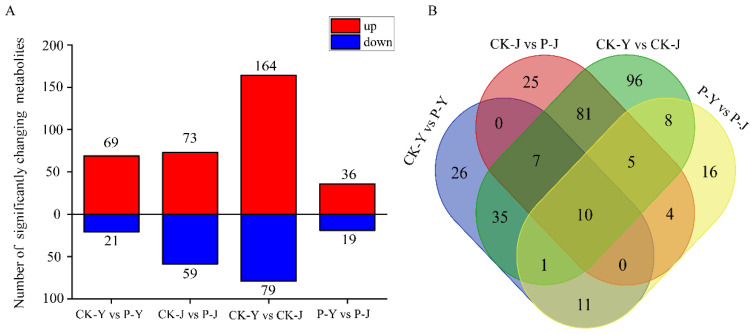
Significantly changing metabolites in response to the compound material regulating the saline and alkaline stresses. (**A**) Number of significantly changing metabolites in cotton leaves at different treatments. (**B**) Venn diagrams of significantly changing metabolites among saline and alkaline stresses in cotton leaves.

**Figure 5 plants-14-00394-f005:**
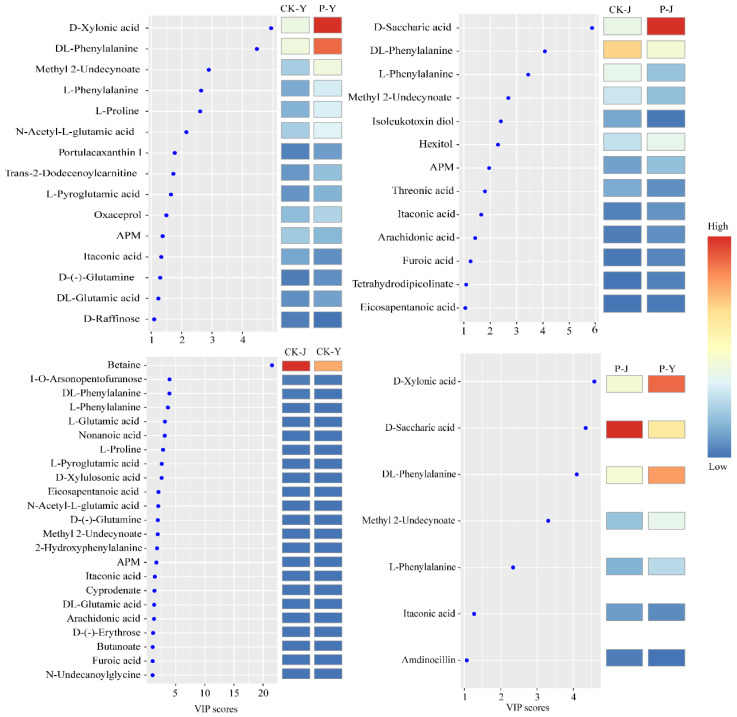
Selected metabolites in response to the compound material regulate saline and alkaline stresses. VIP (variable importance in projection) score of metabolites are shown. Colored boxes indicate the relative contents of the corresponding metabolite in each group.

**Figure 6 plants-14-00394-f006:**
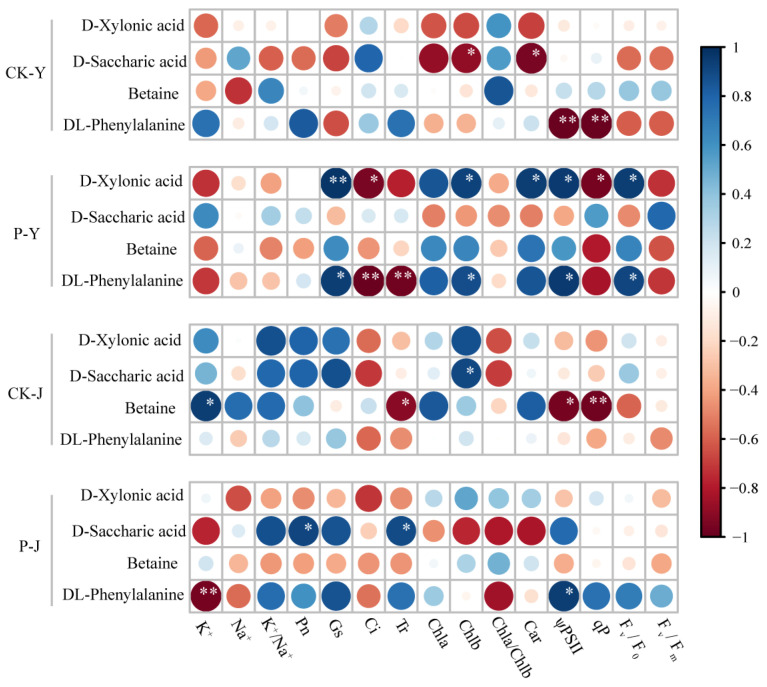
Correlation analysis between selected metabolites and K^+^, Na^+^, and the photosynthesis of cotton leaves. * *p* < 0.05, ** *p* < 0.01. Size of dots indicate the magnitude of the r (correlation coefficient).

**Figure 7 plants-14-00394-f007:**
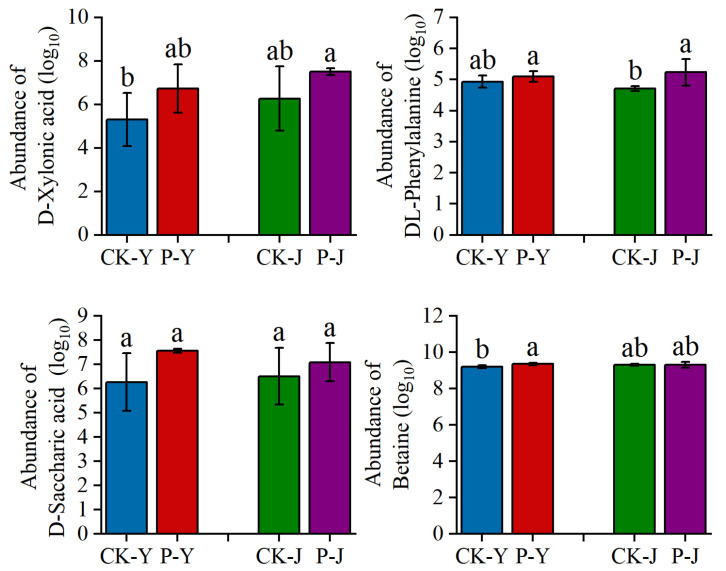
Histogram presentation of some differentially accumulated metabolites in cotton. The *x*-axes on the graphs represent the treatments, while the *y*-axes represent the log_10_ value of the area under the MS peaks. Note: different letters in the bar chart indicate significant differences (*p* < 0.05); the same below.

**Figure 8 plants-14-00394-f008:**
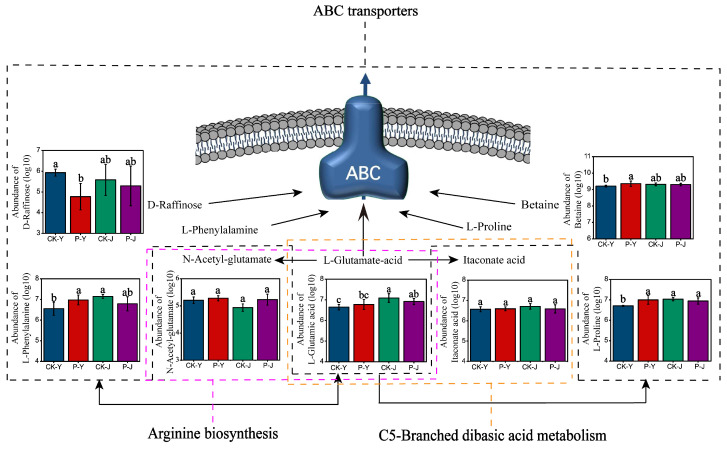
Regulation of metabolic pathway responses to saline–alkaline stress in cotton leaves. Note: different letters in the bar chart indicate significant differences (*p* < 0.05); the same below.

**Figure 9 plants-14-00394-f009:**
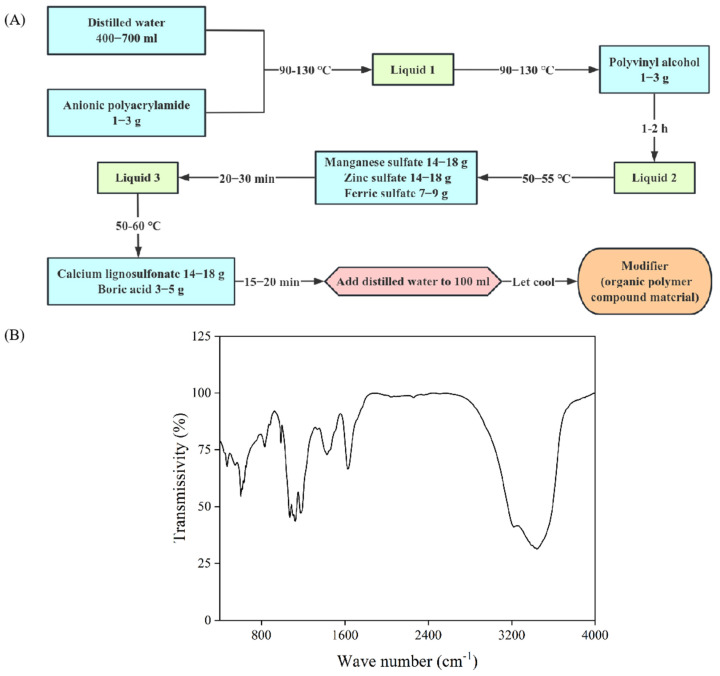
Compound material properties. (**A**) Flowchart; (**B**) Fourier infrared spectra.

**Table 1 plants-14-00394-t001:** Fold changes and *p*-value of selected metabolites in cotton leaves.

Group of Compounds	Metabolite Name	CK-Y vs. P-Y		CK-J vs. P-J		CK-Y vs. CK-J		P-Y vs. P-J	
		Log2(FC)	*p*-Value	Log2(FC)	*p*-Value	Log2(FC)	*p*-Value	Log2(FC)	*p*-Value
Carbohydrates	D-raffinose	−2.83	1.4 × 10^−3^	-	-	-	-	-	-
	D-xylonic acid	1.34	3.2 × 10^−2^	-	-	-	-	−1.04	7.3 × 10^−2^
	D-saccharic acid	-	-	1.34	4.4 × 10^−2^	-	-	0.72	6.2 × 10^−2^
	Threonic acid	-	-	−0.8	6.5 × 10^−2^	-	-	-	-
	Hexitol	-	-	0.42	6.6 × 10^−2^	-	-	-	-
	D-(-)-Erythrose	-	-	-	-	0.66	1.3 × 10^−2^	-	-
	1-O-Arsonopentofuranose	-	-	-	-	0.51	5.0 × 10^−2^	-	-
	D-Xylulosonic acid	-	-	-	-	−3.98	8.7 × 10^−2^	-	-
Amino acids	L-Pyroglutamic acid	0.6	3.00 × 10^−2^	-	-	1.73	1.10 × 10^−2^	-	-
	DL-glutamic acid	0.62	4.00 × 10^−2^	-	-	0.89	9.30 × 10^−2^	-	-
	N-Acetyl-L-glutamic acid	0.64	4.20 × 10^−2^	-	-	0.73	8.00 × 10^−2^	-	-
	DL-phenylalanine	1.11	2.50 × 10^−2^	−0.61	8.40 × 10^−3^	0.89	8.70 × 10^−4^	−0.83	5.90 × 10^−2^
	L-proline	1.11	4.50 × 10^−2^	-	-	1.11	1.20 × 10^−4^	-	-
	D-(-)-Glutamine	1.3	4.00 × 10^−2^	-	-	2.18	1.70 × 10^−2^	-	-
	Portulacaxanthin I	1.36	8.50 × 10^−4^	-	-	-	-	-	-
	APM	−0.43	7.30 × 10^−2^	0.67	6.80 × 10^−2^	-0.65	1.20 × 10^−2^	-	-
	Oxaceprol	0.47	9.00 × 10^−2^	-	-	-	-	-	-
	L-phenylalanine	1.31	5.90 × 10^−2^	−0.94	1.70 × 10^−2^	0.59	1.50 × 10^−2^	−0.82	5.60 × 10^−2^
	Tetrahydrodipicolinate	-	-	0.93	8.90 × 10^−2^	-	-	-	-
	betaine	-	-	-	-	0.38	4.90 × 10^−2^	-	-
	N-Undecanoylglycine	-	-	-	-	0.8	5.00 × 10^−2^	-	-
	2-Hydroxyphenylalanine	-	-	-	-	1.18	4.00 × 10^−2^	-	-
	L-glutamic acid	-	-	-	-	1.54	9.40 × 10^−3^	-	-
	Amdinocillin	-	-	-	-	-	-	4.86	5.60 × 10^−2^
Fatty acids	Methyl 2-Undecynoate	0.91	2.00 × 10^−2^	−0.72	3.20 × 10^−2^	0.62	5.70 × 10^−2^	−1.01	1.30 × 10^−2^
	Itaconic acid	−0.82	6.90 × 10^−2^	0.95	3.00 × 10^−2^	−1.08	2.70 × 10^−2^	0.7	9.10 × 10^−2^
	Trans-2-Dodecenoylcarnitine	1	9.50 × 10^−2^	-	-	-	-	-	-
	Isoleukotoxin diol	-	-	−1.86	3.60 × 10^−2^	-	-	-	-
	Eicosapentanoic acid	-	-	1.04	2.20 × 10^−2^	−1.27	5.80 × 10^−3^	-	-
	Arachidonic acid	-	-	1.29	3.70 × 10^−2^	−1.24	1.80 × 10^−3^	-	-
	Furoic acid	-	-	0.86	5.40 × 10^−2^	−0.85	1.20 × 10^−2^	-	-
	Butanoate	-	-	-	-	0.59	3.70 × 10^−2^	-	-
	Cyprodenate	-	-	-	-	0.68	1.10 × 10^−3^	-	-
	Nonanoic acid	-	-	-	-	0.96	3.10 × 10^−2^	-	-

## Data Availability

The datasets generated during and/or analyzed during the current study are available from the corresponding author on reasonable request.

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
