# Peer review of "Metabolic and Photosynthesis Analysis of Compound-Material-Mediated Saline and Alkaline Stress Tolerance in Cotton Leaves"

_plants, 2025, doi:10.3390/plants14030394_

Round 1
Reviewer 1 Report
Comments and Suggestions for Authors
What is the main question addressed by the research?
A mixture of calcium lignosulfonate, manganese sulfate, zinc sulfate, ferric sulfate, boric acid, anionic polyacrylamide, and polyvinyl alcohol named “modifier” was tested for its ability to alleviate sodium salt stress and alkaline stress in cotton. Moreover, the aim was to determine the photosynthetic and metabolic mechanisms by which the modifier regulates cotton stress tolerance.
What parts do you consider original or relevant for the field? What specific gap in the field does the paper address?
The authors for the first time applied the selected modifier to increase cotton resistance to saline and alkaline stresses and identified key metabolites and metabolic pathways involved in the regulatory mechanism of cotton photosynthesis response to saline and alkaline stresses mediated by the selected modulator.
What does it add to the subject area compared with other published material?
This paper contains new information on the major metabolites and metabolic pathways in cotton leaves under salt and alkaline stress and evidence of stress mitigation by the addition of selected modifier.
What specific improvements should the authors consider regarding the methodology? What further controls should be considered?
My area of research and competence is far from the subject of this paper. Therefore, I find it difficult to suggest specific improvements and further controls regarding the methodology of this study.
Please describe how the conclusions are or are not consistent with the evidence and arguments presented. Please also indicate if all main questions posed were addressed and by which specific experiments.
The conclusions show different effectiveness of the modifier under sodium salt stress and alkaline stress in cotton (Figure 1, 2). In the presence of the modifier, metabolites are significantly changed under saline and alkaline stresses (Figure 4). Key metabolic pathways involved in cotton leaf photosynthesis under saline stress, alkaline stress, and modifier treatment are described in Section 2.6 and summarized in Figure 8. The conclusions emphasize the differences in the metabolic pathways involved in cotton leaf photosynthesis under saline stress, alkaline stress, and the effect of modifier addition.
Are the references appropriate?
The manuscript material uses a good selection of references on the topic of this paper. I did not find any references not related to the topic of the paper at all.
Please include any additional comments on the tables and figures and quality of the data.
Insert before the text on Line 88:
“Hereinafter used: CK-Y or CK-Y treatment (NaCl was applied to soil); P-Y or P-Y treatment (NaCl and modifier were applied to soil); CK-J or CK-J treatment (Na2CO3 was applied to soil); P-J or P-J treatment (Na2CO3 and modifier were applied to soil).”
In the absence of such an explanation, it is difficult to read the text of the publication for the first time.
Author Response
Comment 1: What is the main question addressed by the research?
A mixture of calcium lignosulfonate, manganese sulfate, zinc sulfate, ferric sulfate, boric acid, anionic polyacrylamide, and polyvinyl alcohol named “modifier” was tested for its ability to alleviate sodium salt stress and alkaline stress in cotton. Moreover, the aim was to determine the photosynthetic and metabolic mechanisms by which the modifier regulates cotton stress tolerance.
Response 1: Thanks so much for reviewing and evaluating the manuscript. We greatly appreciate your positive feedback and thoughtful comments.
Comment 2: What parts do you consider original or relevant for the field? What specific gap in the field does the paper address?
The authors for the first time applied the selected modifier to increase cotton resistance to saline and alkaline stresses and identified key metabolites and metabolic pathways involved in the regulatory mechanism of cotton photosynthesis response to saline and alkaline stresses mediated by the selected modulator.
Response 2: Thanks so much for reviewing and evaluating the manuscript. We greatly appreciate your positive feedback and thoughtful comments.
Comment 3: What does it add to the subject area compared with other published material?
This paper contains new information on the major metabolites and metabolic pathways in cotton leaves under salt and alkaline stress and evidence of stress mitigation by the addition of selected modifier.
Response 3: Thanks so much for reviewing and evaluating the manuscript. We greatly appreciate your positive feedback and thoughtful comments.
Comment 4: What specific improvements should the authors consider regarding the methodology? What further controls should be considered?
My area of research and competence is far from the subject of this paper. Therefore, I find it difficult to suggest specific improvements and further controls regarding the methodology of this study.
Response 4: Thanks so much for reviewing and evaluating the manuscript. We greatly appreciate your positive feedback and thoughtful comments.
Comment 5: Please describe how the conclusions are or are not consistent with the evidence and arguments presented. Please also indicate if all main questions posed were addressed and by which specific experiments.
The conclusions show different effectiveness of the modifier under sodium salt stress and alkaline stress in cotton (Figure 1, 2). In the presence of the modifier, metabolites are significantly changed under saline and alkaline stresses (Figure 4). Key metabolic pathways involved in cotton leaf photosynthesis under saline stress, alkaline stress, and modifier treatment are described in Section 2.6 and summarized in Figure 8. The conclusions emphasize the differences in the metabolic pathways involved in cotton leaf photosynthesis under saline stress, alkaline stress, and the effect of modifier addition.
Response 5: Thanks so much for reviewing and evaluating the manuscript. We greatly appreciate your positive feedback and thoughtful comments.
Comment 6: Are the references appropriate?
The manuscript material uses a good selection of references on the topic of this paper. I did not find any references not related to the topic of the paper at all.
Response 6: Thanks so much for reviewing and evaluating the manuscript. We greatly appreciate your positive feedback and thoughtful comments.
Comment 7: Please include any additional comments on the tables and figures and quality of the data.
Insert before the text on Line 88:
“Hereinafter used: CK-Y or CK-Y treatment (NaCl was applied to soil); P-Y or P-Y treatment (NaCl and modifier were applied to soil); CK-J or CK-J treatment (Na2CO3 was applied to soil); P-J or P-J treatment (Na2CO3 and modifier were applied to soil).”
In the absence of such an explanation, it is difficult to read the text of the publication for the first time.
Response 7: Thanks for your questions. We inserted four treatments in Introduction. Please see lines 78-81.
“In this field barrel experiment, four treatments were designed, including (1) CK-Y: NaCl was applied to soil, (2) P-Y: NaCl and compound material were applied to soil, (3) CK-J: Na2CO3 was applied to soil, and (4) P-J: Na2CO3 and compound material were applied to soil. ”

Reviewer 2 Report
Comments and Suggestions for Authors
See the attached file

Comments on the Quality of English Language
See the Review file
Author Response
Reviewer 2:
General comments
This ms. presents the results of an experiment in which cotton plants were grown in soil and subjected to NaCl and Na2SO4 stress, and to addition of a mixture of organic compounds called “the modifier”. Measurements of the content of Na, K, plus metabolome analysis as well as measurements of photosynthesis-related physiological parameters showed the response of the plants to the various treatments. Addition of the modifier relieved some of the effects of the stress to different degrees, and a possible biochemical mode of its action is discussed. This is a very thorough and detailed report of novel and interesting data.
Comment 1: Certain controls are missing parts in the experimental design: A. Control – No Stress, No Modifier, and B. Control – No Stress + Modifier. In the absence of these controls, it is impossible to assess the magnitude of the stress and of the modifier effect. Additionally, it would be important to know the effect of the modifier on the SOIL of the control and stress treatment (ion availability, water availability, and pH).
Response 1: Thanks for your questions. Unfortunately, Control – No Stress + Modifier was not designed in this study. However, in Materials and Methods, the Fourier Transform Infrared Spectroscopy (FTIR) (Figure 9B) showed that the composition of the compound material was complex. The peak absorption at 3219 cm-1 and 1620 cm-1 could be attributed to the N-H stretching vibration and bending vibration of primary amine, respectively, the peak absorption at 1466 cm-1 could be attributed to the stretching vibration of aromatic ring C=C, the peak absorption at 1145 cm-1 could be attributed to the symmetrical stretching vibration of PO2 in inorganic phosphate, the peak absorption at 823 cm-1 could be attributed to the =C-H non-planar bending vibration on the benzene ring, and the peak absorption at 545 cm-1 could be attributed to the vibration of metal oxides (Figure 9B). Besides, this study only focused on the binding of soil saline and alkaline stresses.
The purpose of this experiment was to compare the effects of compound material on the photosynthetic characteristics and metabolites of cotton under saline and alkaline stresses.
Cotton was sown on May 4. Six seedlings were retained in each barrel after cotton emergence. At the flowering and boll-forming stage (August 19th), five new cotton leaves (top leaves) with similar growth status were randomly selected from each treatment in the full flowering stage for metabolic and photosynthesis analysis, and the soil physical and chemical properties of each treatment were determined (Table S1).
Table S1. Effects of compound material on soil physical and chemical properties under saline and alkaline stresses.
Treatment |
K+ content (mg·g-1) |
Na+ content (mg·g-1) |
pH |
EC value (us·cm-1) |
CK-Y |
1.40±0.02 a |
5.00±0.06 c |
8.02±0.11 c |
24.10±1.35 abc |
P-Y |
1.07±0.07 b |
4.49±0.35 c |
7.76±0.07 d |
22.05±0.57 c |
CK-J |
1.41±0.03 a |
8.52±0.17 a |
9.85±0.04 a |
27.65±1.48 a |
P-J |
1.16±0.03 b |
5.75±0.35 b |
9.32±0.11 b |
23.65±1.48 bc |
Comment 2: I wonder if this is intended to applicable in field scale or just a curiosity for potted plants. The paragraph in lines 159-175 and Figure 6 contain too much details and are very difficult to follow. I suggest that only correlations that are considered important or meaningful will be presented here and the full set of data will be presented as on-line supporting material.
Response 2: Thanks for your questions. Our experiment was carried out field scale. “On April 20th, cotton field soil was collected by layers and transferred into plastic barrels (0.5 m in diameter and 0.6 m in height) while keeping the original soil layer arrangement and bulk density. Then, the barrels were buried back to a cotton field.” Please see lines 376-378. The barrels were buried in the field and the original soil profile was maintained.
According to your suggestion, the correlation analysis of p<0.05 were removed, and the correlation analysis of p<0.01 in Figure 6 were presented. Please see lines 168-175.
Specific comments
Comment 3: A list of abbreviations at the top of the article can improve the clarity of the text including: CK-Y, CKJ, P-Y, P-J, Car, Gs, Pn, Tr, VIP, DAMs, PSII, qP,Ci, ABC, EC
Response 3: Thanks for your questions. According to the journal format requirements, a list of abbreviations was added before the References. Please see lines 486-504.
Comment 4: Caption of Figure 6: Explain the meaning of the asterisks and the different size of dots.
Response 4: Thanks for your questions. We have modified the title of Figure 6.
“Figure 6. Correlation analysis between selected metabolites and K+, Na+, photosynthesis of cotton leaves. * P < 0.05, ** P < 0.01. Size of dots indicate the magnitude of the r (correlation coefficient).”
Comment 5: Lines 189-192: Note that Figure 7 shows NO significant differences among the columns regarding the D-Saccharic acid level in the four treatments. The text should reflect that correctly.
Response 5: Thanks for your questions. The text description was modified, and metabolites were changed between treatments but no significance. Please see lines 188-194.
Comment 6: Line 222: the statement: “alkaline stress … increases the pH in plants” is incorrect. The pH change in the soil does not cause a direct similar change within the plant tissues.
Response 6: Thanks for your questions. We have modified the statement. Please see lines 221-224.
“This is due to the fact that saline stress generally leads to ionic damage and osmotic stress in plants. However, high pH in alkaline stress directly affects plant roots growth and indirectly affects plants uptake of soil nutrients [17]. ”
Comment 7: Lines 257-259: I don’t see the basis or the reason in the statement: “The improvement of photosynthetic characteristics is due to the excellent water retention capacity of the polymer materials in modifier, which could decrease the water evaporation in saline and alkaline soils.“
Response 7: Thanks for your questions. We have added a relevant reference.
Ahmed, S.; Islam, M.S.; Antu, U.B.; Islam, M.M.; Rajput, V.D.; Mahiddin, N.A.; Paul, J.R.; Ismail, Z.; Ibrahim, K.A.; Idris, A.M. Nanocellulose: A novel pathway to sustainable agriculture, environmental protection, and circular bioeconomy. Int. J. Biol. Macromol. 2025, 285, 137979.
Comment 8: Lines 269-274: This text mixes cause and effect of the betaine abundance. It starts by saying “Under NaCl stress the increase in betaine can improve the stomatal characteristics…” but ends saying that “… stress deceased the Tr, PSII, and qP, leading to an increase in betaine abundance.”
Response 8: Thanks for your questions. We have modified the statement. Please see lines 274-281.
“Hamani et al. have shown that the increase in betaine can effectively improve the stomatal characteristics of cotton seedlings, thereby rapidly improving the gas exchange and chlorophyll fluorescence of cotton seedling under NaCl stress [20]. In this study, the abundance of betaine of cotton leaves was higher under Na2CO3 stress than under NaCl stress. This may be due to that Na2CO3 stress decreases the Tr, ψPSII, and qP, leading to an increase in betaine abundance (Figure 7). The different results may be due to the difference in metabolites between the seedling stage and the flowering and boll stage.”
Comment 9: Lines 278-281: I do not see the basis for the statements that describe the changes in organic acids content being the causes of the improvement of photosynthesis parameters.
Response 9: Thanks for your questions. The results were described based on correlation analysis (Figure 6) and abundance levels of key metabolites (Betaine, D-xylonic acid, DL-phenylalanine, and D-saccharic acid) (Figure 7). Please see lines 284-289.
“In this study, under NaCl stress, the photosynthetic characteristics (Gs, Chlb, ψPSII, and Fv/Fo) of cotton leaves were improved through increasing the abundance of D-Xylonic acid and DL-Phenylalanine (Figure 6 and Figure 7). Under Na2CO3 stress, the application of compound material improved the Pn and Tr by increasing the abundance of D-Saccharic acid (Figure 6 and Figure 7).”
Comment 10: Line 304: I do not see the basis for the statement “The accumulation of amino acids could provide sufficient energy for ABC transporters.”
Response 10: Thanks for your questions. We have added a relevant reference.
Ingrisano, R.; Tosato, E.; Trost, P.; Gurrieri, L.; Sparla, F. Proline, cysteine and branched-chain amino acids in abiotic stress response of land plants and microalgae. Plants 2023, 12(19), 3410.
Comment 11: Lines 313-315: I do not see the basis for the following statement “..less output of betaine by the ABCtransporters…indicating that the damage of saline stress … was less than that of alkaline stress”.
Response 11: Thanks for your questions. We have modified the statement. Please see lines 319-325.
“Betaine can regulate the high osmotic pressure through its transport and accumulation by ABC transporters, and play a protective role against saline stress [39]. In this study, the abundance of betaine was significantly increased by compound material under saline stress, but was decreased under alkaline stress (Figure 8). This may be due to the less output of betaine by ABC transporters pathway under saline stress, indicating that the damage of saline stress to leaves was less than that of alkaline stress. ”
Comment 12: Lines 342-344: No explanation is given to the reason behind the composition of the modifier, nor there was one in the previous article by the same group (reference 41).
Response 12: Thanks for your questions. The compound material used in this study is based on the soil testing for formulated fertilization and tailored to local conditions through independent research.
Comment 13: Line 369: I guess that the measurements of “soil pH and electrical conductivity” were done after dilution with a certain amount of water (1:5?). Please specify.
Response 13: Thanks for your questions. We have added determination of soil pH and EC. Please see lines 381-383.
“The soil pH and electrical conductivity (EC) (water: soil=2.5:1) were 8.24 and 18.4 ds·m-1, respectively, for NaCl treatments, and 9.78 and 10.3 ds·m-1, respectively, for Na2CO3 treatments.”
Comment 14: Lines 423, 428: Replace “America” by “Thermo Fisher Scientific”.
Response 14: Thanks for your questions. We have replaced. Please see lines 437, 442.
Language
In general, the English is well written but see below a umber of points to be considered
Comment 15: Line 31: replace “metabolismye” by “metabolism. Response 15: Thanks for your questions. We have replaced. Please see lines 33.
Comment 16: Line 95: Why are the “modifiers” referred to as “compound material” a term which is not used anywhere else in the text?
Response 16: Thanks for your questions. We have unified the name of the organic polymer compound material as “compound material” following your suggestion.
Comment 17: Line 116: What do they mean by “organic oxygen compounds”?
Response 17: Thanks for your questions. Organic oxygen compounds mainly include carbonyl compounds, carbohydrates and carbohydrate conjugates, alcohols and polyols, and ethers.
Comment 18: Line 122: Add “of” to read “comparison of group…” Response 18: Thanks for your questions. We have replaced. Please see lines 131.
Comment 19: Line 252: Replace “after” by “under”. Response 19: Thanks for your questions. We have replaced. Please see lines 254.
Comment 20: Line 263: Replace “Modifier by “The modifier”. Response 20: Thanks for your questions. We have replaced. Please see lines 267.
Comment 21: Line 266: Replace “causes” by “caused”. Response21: Thanks for your questions. We have replaced. Please see lines 270.
Comment 22: Line 277: Replace “regulated” by “affected”. Response 22: Thanks for your questions. We have replaced. Please see lines 284.
Comment 23: Line 291: Replace “generating” by “generated”. Response 23: Thanks for your questions. We have replaced. Please see lines 299.
Comment 24: Lines 311 and 314: Replace “in” by “by”. Response24: Thanks for your questions. We have replaced. Please see lines 320, 323.
Comment 25: Line 328: Replace “due to that modifier alleviate” by “due to the alleviation of … by the modifier”. Response 25: Thanks for your questions. We have replaced. Please see lines 339.
Comment 26: Line 438: replace “draw” by “drawn”. Response 26: Thanks for your questions. We have replaced. Please see lines 452.
Comment 27: Lines 447, 449: Replace “to improve” by “improved”.
Response 27: Thanks for your questions. We have replaced. Please see lines 464, 466.

Reviewer 3 Report
Comments and Suggestions for Authors
I have read the manuscript carefully and find the topic compelling.
The study addresses an important issue: salinity stress, a major environmental challenge that negatively impacts crop development.
Soil salinization leads to saline and alkaline stress, primarily due to high concentrations of sodium salts, including neutral salts, and alkaline salts. Prior studies often used NaCl to represent neutral salts and Na2CO3 for alkaline salts. The effects of these salts on crop growth vary significantly; neutral salt stress mainly causes ion toxicity and osmotic pressure, while alkaline salt stress inflicts similar damage while also elevating pH levels within plants.
This study aims to: (i) examine how modifiers affect the photosynthetic traits and metabolic profiles of cotton under saline and alkaline stress, and (ii) identify key metabolites and metabolic pathways involved in cotton's photosynthetic response under these conditions.
Ultimately, this research seeks to provide a technical guide for improving cotton tolerance to saline and alkaline stress in arid and semi-arid regions.
However, there are several areas for improvement:
>The manuscript maybe requires enhancements to elevate overall quality of paper.
>Focus on clarifying and enriching the abstract and discussion sections with relevant details, supporting evidence.
>In the concluding remarks, emphasize the novelty of the research findings.
>The literature survey needs expansion. Incorporate 4-6 additional relevant studies from this Journal, particularly avoiding self-citation.
In addition, please do not ignore works from Norio Murata's group on salt stress.
Overall, please try to condense the text to improve readability while ensuring that all necessary information is retained
Comments on the Quality of English Language
I think it is OK.
Author Response
Reviewer 3:
Comments and Suggestions for Authors
I have read the manuscript carefully and find the topic compelling.
The study addresses an important issue: salinity stress, a major environmental challenge that negatively impacts crop development.
Soil salinization leads to saline and alkaline stress, primarily due to high concentrations of sodium salts, including neutral salts, and alkaline salts. Prior studies often used NaCl to represent neutral salts and Na2CO3 for alkaline salts. The effects of these salts on crop growth vary significantly; neutral salt stress mainly causes ion toxicity and osmotic pressure, while alkaline salt stress inflicts similar damage while also elevating pH levels within plants.
This study aims to: (i) examine how modifiers affect the photosynthetic traits and metabolic profiles of cotton under saline and alkaline stress, and (ii) identify key metabolites and metabolic pathways involved in cotton's photosynthetic response under these conditions.
Ultimately, this research seeks to provide a technical guide for improving cotton tolerance to saline and alkaline stress in arid and semi-arid regions.
However, there are several areas for improvement:
Comment 1: The manuscript maybe requires enhancements to elevate overall quality of paper.
Response 1: Thanks for your questions. We have thoroughly checked the manuscript, and corrected some typos errors.
Comment 2: Focus on clarifying and enriching the abstract and discussion sections with relevant details, supporting evidence.
Response 2: Thanks for your questions. We have modified the abstract and discussion sections following your suggestion.
Comment 3: In the concluding remarks, emphasize the novelty of the research findings.
Response 3: Thanks for your questions. We have added the novelty following your suggestion. Please see lines 458-460.
“The first time applied the organic polymer compound material to increase cotton resistance to saline and alkaline stresses and identified key metabolites involved in the regulatory mechanism of cotton photosynthesis.”
Comment 4: The literature survey needs expansion. Incorporate 4-6 additional relevant studies from this Journal, particularly avoiding self-citation.
Response 4: We have added relevant literature following your suggestion.
Liu, X.; Su, S. Growth and physiological response of Viola tricolor L. to NaCl and NaHCO3 Stress. Plants 2023, 12(1), 178.
Shafiq, H.; Shani, M.Y.; Ashraf, M.Y.; De Mastro, F.; Cocozza, C.; Abbas, S.; Ali, N.; Zaib-Un-Nisa; Tahir, A.; Iqbal, M.; Khan, Z.; Gul, N.; Brunetti, G. Copper oxide nanoparticles induced growth and physio-biochemical changes in maize (Zea mays L.) in saline soil. Plants 2024, 13(8), 1080.
Hitti, Y.; MacPherson, S.; Lefsrud, M. Separate effects of sodium on germination in saline-sodic and alkaline forms at different concentrations. Plants 2023, 12(6), 1234.
Ingrisano, R.; Tosato, E.; Trost, P.; Gurrieri, L.; Sparla, F. Proline, cysteine and branched-chain amino acids in abiotic stress response of land plants and microalgae. Plants 2023, 12(19), 3410.
Comment 5: In addition, please do not ignore works from Norio Murata's group on salt stress.
Response 5: We have added relevant literature following your suggestion.
Allakhverdiev, S.I.; Murata, N. Salt stress inhibits photosystems II and I in cyanobacteria. Photosynth. Res. 2008, 98(1-3), 529-539.
Murata, N.; Nishiyama, Y. ATP is a driving force in the repair of photosystem II during photoinhibition. Plant. Cell. Environ. 2018, 41(2), 285-299.
Comment 6: Overall, please try to condense the text to improve readability while ensuring that all necessary information is retained
Response 6: Thanks for your questions. We have thoroughly condensed the manuscript, and retained all necessary information.
Comment 7: Comments on the Quality of English Language
I think it is OK.
Response 7: Thanks so much for reviewing and evaluating the manuscript.

Round 2
Reviewer 3 Report
Comments and Suggestions for Authors
Thnak you.